# Ecological Interactions among Thrips, Soybean Plants, and Soybean Vein Necrosis Virus in Pennsylvania, USA

**DOI:** 10.3390/v15081766

**Published:** 2023-08-18

**Authors:** Asifa Hameed, Cristina Rosa, Cheryle A. O’Donnell, Edwin G. Rajotte

**Affiliations:** 1Department of Entomology, Pennsylvania State University, State College, PA 16802, USA; uvu@psu.edu; 2Plant Pathology and Environmental Microbiology, Pennsylvania State University, State College, PA 16802, USA; czr2@psu.edu; 3USDA APHIS PPQ National Identification Services National Specialist (Thysanoptera and Psylloidea), Systematic Entomology Laboratory, B-005, Rm 137 BARC-West, 10300 Baltimore Avenue, Beltsville, MD 20705, USA; cheryle.a.odonnell@aphis.usda.gov

**Keywords:** thrips fauna, protein content, fiber content, oil content, population dynamics, photosynthesis rate, stomatal conductance, carbon dioxide exchange, environmental variables, overwintering/hibernation behavior

## Abstract

Analysis of ecological and evolutionary aspects leading to durability of resistance in soybean cultivars against species *Soybean vein necrosis orthotospovirus* (SVNV) (Bunyavirales: Tospoviridae) is important for the establishment of integrated pest management (IPM) across the United States, which is a leading exporter of soybeans in the world. SVNV is a seed- and thrips- (vector)-borne plant virus known from the USA and Canada to Egypt. We monitored the resistance of soybean cultivars against SVNV, surveyed thrips species on various crops including soybeans in Pennsylvania, and studied thrips overwintering hibernation behavior under field conditions. Field and lab experiments determined disease incidence and vector abundance in soybean genotypes. The impact of the virus, vector, and their combination on soybean physiology was also evaluated. Seed protein, fiber, oil, and carbohydrate content were analyzed using near infra-red spectroscopy. We found that the variety Channel3917R2x had higher numbers of thrips; hence, it was categorized as preferred, while results showed that no variety was immune to SVNV. We found that thrips infestation alone or in combination with SVNV infection negatively impacted soybean growth and physiological processes.

## 1. Introduction

Soybean is a highly valued leguminous source of oil and protein [1]. Many of the world’s 7.8 billion people rely on this valuable commodity for domestic oil, food, and feed purposes [2]. Soybean meal is important for livestock, poultry, and aquaculture [3]. Brazil, the United States, and Argentina dominate in soybean production [4]. In 2022, the United States produced 4.3 billion bushels of soybeans, with Pennsylvania producing 29.89 million of these bushels [5].

Soybean is affected by a number of arthropod pests and diseases. Among the pathogens causing diseases on soybean, the species *Soybean vein necrosis orthotospovirus* (SVNV) (Orthotospovirus: Tospoviridae: Bunyavirales) is a widely recognized seed-borne and vector-transmitted virus [6,7,8,9,10,11]. While the seed-borne transmission route is limited, it provides for movement of the virus into new areas through trade routes. This virus is well established in all soybean growing regions of the USA, Canada, and Egypt [7,12,13]. SVNV is transmitted at a very high rate by soybean thrips, *Neohydatothrips variabilis* (beach), but other thrips species (*Frankliniella tritici* and *F. fusca*) can also transmit the virus with reduced transmission efficiency in the United States [9,14,15,16,17]. In Egypt, *Caliothrips phaseoli*, *Megalourothirps sjostedti*, and *F. occidentalis* transmitted SVNV at a transmission efficiency of, respectively, 6.7, 3.3, and 3.4% under laboratory conditions [13].

Soybean is often grown in close proximity to other field crops and wild plant species [10]. Thrips can inhabit a wide range of plants, and various plants can act as reservoir host plants of both the vector and virus [11,18]. Information about SVNV and thrips alternative hosts is lacking, but laboratory studies have provided some insight into possible alternative host plants of the virus and vectors [7,13,19,20].

One theory about soybean thrips is that these thrips migrate each year from south to north on storm fronts in the summer [21], however, recent studies on alternative host plants of soybean in greenhouse conditions documented that in the mid-west U.S., the soybean thrips might not migrate at all. Instead, they may actually overwinter on perennials and in the spring migrate to cover crops and then to soybeans [22]. Bloomingdale et al. [22] released thrips on different plants, including cover crops, and observed larval and adult stages feeding on some of these plants [22], and hypothesized that the thrips overwinter on them. Although these studies were intriguing, thrips face different challenges in the winter to survive, including cold weather, heavy frosts, and lack of food [23]. Thrips may survive either through behavioral freeze avoidance or tolerance [24]. It is currently unknown whether thrips migrate from the north to the south for survival, but several species are known to overwinter as adults, either on vegetation or in the soil [24,25,26]. However, soybean thrips are not known to overwinter in northern US states [21].

For integrated pest management (IPM), it is important to understand the primary disease spread route, i.e., understand the ecology of the vectors which maintain the inoculum and their timing of invasion on the principal crop as well as the relationships with alternative hosts where the vectors feed before arriving on the principle crops [16,27]. Some alternative host crops can act either as open ended hosts, which maintain the thrips and virus, or dead end hosts, which can maintain the virus but cannot support thrips [28]. Subsequently, it is also important to understand secondary spread behavior and the factors which lead to dispersal from primary vector population abundance, which includes weather factors and crop phenology. Chitturi et al. [29] monitored soybean thrips (*N. variabilis*) populations using sticky traps in Alabama, and found that the population of soybean thrips peaked in the third week of June in different counties (Tallassee, Auburn and Headland) in 2015, while in 2016 the maximum population of thrips were observed in first week of July in Tallassee, Auburn and Headland. In Indiana, Keough [14] observed higher *N. variabilis* numbers in the month of August on soybean crops. A study was conducted to determine effect of soybean vein necrosis virus and thrips infestation on yield reduction in soybeans planted under greenhouse conditions [30]. The study concluded that plants infested with thrips and SVNV at the V1 stage died early and did not produce seeds or pods [30].

Virus infection and vector feeding may cause serious decreases in yield [31]. Plant varieties may be more or less at risk from SVNV depending upon vector abundance [32]. It is important to select plant cultivars which have defenses against the herbivore, but also have good seed qualitative traits (seed protein, oil, fiber, and carbohydrate content).

We studied thrips species’ composition and hibernation behavior, the relative abundance of thrips on different soybean varieties, seed quality, the effect of weather on thrips abundance, the impact of the virus and vector pairing on grain quality parameters, and the impact of the virus and vector on the physiology of soybean in central Pennsylvania under lab and field conditions. This is the first field study describing the resistance potential of soybean cultivars to thrips abundance under field conditions, as well as the first study to investigate the effect of SVNV on plant physiology. The knowledge gained through this research could be useful for the development of IPM strategies against vector and virus dispersal.

## 2. Materials and Methods

### 2.1. Soybean Field Layout

During 2016 and again in 2017, the relative abundance of thrips on 10 soybean varieties was determined in the field. Cultivars were planted in a randomized complete block design (RCBD) at the Rock Spring Russel E. Larson Agricultural Research Facility in PA. Two rows were left uncultivated to maintain a buffer between varieties. The plot size was 16 × 5.2 square feet with four replicates. A two-square-foot wide strip was left as buffer between each plot. Standard tillage and agronomic practices for soybean were used. Ten soybean varieties were planted, including Sway SG3322 (V1) (Seed way, New York, NY, USA), GrowMark FS (Hisoy HS39T60) (V2) (Growmark FS, LLC, Seaford, DE, USA), Grow Mark FS Hisoy HS30A-42 (V3) (Growmark FS, LLC, Seaford, DE, USA), Mycogen 5N343R2 (V4) (Dow Agro Sciences, Calgary, AB, Canada), H3h-12R2 (V5) (Hubner seeds, St. Louis, MO, USA), Hubner3917R2x (V6) (Hubner seeds, USA), Syngenta S27-J7 (V7) (Syngenta, Greensboro, NC, USA), Seed way SG3555 (V8) (Seed way, New York, NY, USA), Mycogen 5N312R2 (V9) (Dow Agro Sciences, Calgary, AB, Canada), and Syngenta NKS36Y6 (V10) (Syngenta, Greensboro, NC, USA). No irrigation, fertilizer, herbicide, or insecticide were applied during the growing season. Seeds were planted using a research soybean plot planter (Wintersteiger, Ried im Innkreis, Austria) with 7-inch row spacing, with 9 rows per plot, 10 plots, and 432 seeds planted per plot. Seeds were inoculated with *Bradyrhizobium japonicum* (S type inoculant, Hancock seed company, Dade City, FL, USA) before planting at the rate of 15 oz per 300 lbs of soybean seeds. The planting depth was 4 cm. The row direction in both years was east to west. The previous planted crop was corn in both years. The land was prepared using a conventional tillage system (two ploughings followed by planking before cultivation). Soybeans were planted in July and harvested in November each year. Harvesting using a Wintersteiger Nursery Master Combine (Wintersteiger, Ried im Innkreis, Austria) was carried out each year in November, when the moisture content in seeds was less than 12%, measured by the PQ-520 Single Kernel Grain Moisture Tester (Kett Electric Laboratory, Tokyo, Japan).

#### 2.1.1. Thrips Abundance

Population dynamics of thrips on the soybean plants was observed by counting thrips on the plant leaves using a hand lens. Five leaves per niche per plant were randomly selected. Niches are the three positions of plant canopy, viz. upper canopy leaves, middle canopy leaves, and lower canopy leaves. A total of 15 leaves per replicate were observed at weekly intervals until crop maturity. Only presence and number of *N. variabilis* (all stages) were recorded to determine thrips abundance from 1 July to 30 September (approximately 12 weeks) in 2016 and 2017.

#### 2.1.2. Determination of Virus Presence in Field Plants from 2016–2017

Five symptomatic leaf samples from each variety were randomly collected for SVNV detection. SVNV presence was determined through ELISA according to the manufacturer’s protocol (Agdia, Elkhart, IN, USA). The plant leaf samples which had an absorbance value measured through an ELISA plate reader (Bio Rad, Hercules, CA, USA) at 405 nm three times or higher than the PBST (phosphate buffer salined with twin 20) control, were considered positive. Although sampling was carried out during 2016, to assess SVNV incidence, all samples were ELISA negative. This might be because the samples were taken from uninfected plants. During 2017, all soybean samples collected during the crop growth stages were negative until symptoms appearance in August; however, after the appearance of symptoms, the samples taken from symptomatic plants were positive through ELISA.

#### 2.1.3. Correlations of Thrips Abundance to Weather Factors during 2016–2017

A regression analysis was conducted to compare *Neohydatothrips variabilis* abundance and the weather factors (air temperature, relative humidity, precipitation, rainfall, and solar radiation). Weather data were obtained from the weather station at the Russel E. Larson Agricultural Research Station. Mean thrips population was plotted against weather factors, viz. temperature, rainfall, relative humidity, wind velocity and solar radiation. Regression and correlation analysis was carried out using R 3.5.3.

#### 2.1.4. Assessment of Grain Quality Parameters

Under field conditions, multiple factors lower the quality of grains, including insects and diseases. Multiple viruses and diseases may lower the crop quality. Hence, to determine the impact of thrips and SVNV incidence on the seed quality, 500 g of seeds were harvested from each variety and each replicate in the field experiment (4 replicates per variety totaling 40 samples per experiment per year) and sent to the Grain Quality Lab at Iowa State University, USA (3167 National Swine Research and Information Center (NSRIC)). The chemical composition of seeds was evaluated through non-destructive near-infra-red analysis. Grain quality parameters (oil content, carbohydrate content, and protein content) were plotted against thrips abundance.

### 2.2. Thrips Species Survey on Soybeans, Field Crops, Weeds, and Ornamentals

Thrips species were collected during summer 2018 on soybeans, weeds, field crops, and ornamentals. Sampling was carried out at multiple locations in Rockspring, PA (40.71; −77.94) at the Russel E. Larson Agricultural Research Station, as well as at the arboretum on the Penn State University Park campus (40.80; −77.86) and at different locations within 5km of the campus. For field crops (soybean, squash, melon, onions, and corn), fifteen plants were randomly checked for thrips presence from each host crop every week in spring, summer, and autumn (from April–November, 2018–2019). For ornamentals (peonies, red clover, nasturtium, white aster daisy, and viburnum), the plants were thoroughly sampled for thrips presence, but most sampling was carried out from inflorescences during spring and summer on weekly basis. In our survey, we did not find many thrips species on trees; however, the bark of trees was checked for thrips species’ presence along with the foliage. The sampling was carried out weekly during summer. The thrips were collected using a beating sheet and then transferred to zip lock bags and brought to the lab. Adults were put in 70% ethanol. For identification, thrips were placed in 10% KOH over-night and then heated in a waterbath for 30 min at 70 °C. Then, thrips were dehydrated in 70% and 95% ethanol, and slide mounted in Canada balsam. Thrips were identified by the species level keys by using Hoddle MS (2012). Species identification and collection was only conducted in 2018.

### 2.3. Effect of SVNV on Plant Physiology

In order to determine whether SVNV affects plant physiology, we conducted an experiment in the green house and growth chamber. For this purpose, we first established a protocol for virus inoculation.

#### 2.3.1. Protocol Establishment for SVNV Inoculation of Virus in Soybean

Because prior attempts of the mechanical inoculation method were unsuccessful [7], we compared various inoculation techniques, such as rubbing the plant leaf surface, syringe inoculation, and thrips inoculation. Viruliferous thrips and virus-infected leaves (taken from the colony of infected thrips) were homogenized by mortar and pestle in disodium phosphate buffer (7.6 pH) plus a pinch of carborundum. Soybean plants at the V2 stage were selected for virus inoculation. The supernatant was used for leaf rub inoculation or injected directly in the leaves using a needleless syringe early in the morning, after keeping the plants in the dark overnight. For thrips transmission 5, 10, and 15 viruliferous thrips were released on individual plants covered by a plastic bottle within a cage. Plants were sprayed with insecticide after 24 h. The infection status of the SVNV-inoculated plants was confirmed through qRT PCR [9] 10 days after the inoculation on new leaves.

#### 2.3.2. Experimental Set-Up for Physiology Experiment

Insects: *N. variabilis* were collected using the beating sheet method during summer 2016 from the Russell E. Larson Agricultural Research Station in State College, Pennsylvania. Live specimens were transferred from aerated jars with soybean leaves for food and placed in a cool box to transfer to the laboratory. Here, the insects were raised on soybean plants in rearing cages (L24.5 × W24.5 × H24.5 cm) inside a growth chamber (Conviron, Winnipeg, MB, Canada) at 25 ± 2 °C and 78–80% RH, LD 14:10.

Plants: Commercial soybean varieties are not always available, so we used seeds from 2017 field season harvested crop, stored at 4 °C. Soybean seeds from three field varieties (SG3322, Channel3917R2X and SG3555) were planted on 28 October 2018 in 4 × 4 × 4 square inch pots in standard potting soil mix (Miracle Gro Potting Soil Mix, Performance Organics, New Hampshire, USA). One gram of Osmocote fertilizer (Osmocote plus Bloomington Brands, LLC, Oxford, PA, USA) was also added at the time of sowing, and no fertilizer was applied afterwards. Each treatment consisted of 3 varieties and 5 individual plants per variety. These varieties were selected on the basis of ELISA results field studies 2017 and average thrips number per year over 2-years of field data. Pots were kept in growth chambers (Conviron, Winnipeg, MB, Canada) at 25 ± 2 °C, L:D 14:10, 78 ± 10% R.H and light flux of 5000 watts. The experiment comprised four different treatments: control (uninfected, untreated/undamaged control plants), mock (plants were injected and rubbed with buffer only), SVNV infection via mechanical inoculation (SVNVMI) (virus was inoculated in the plants using a syringe and rubbing using the first true leaf stage), and SVNV infection via viruliferous thrips and thrips infestation (SVNVIT) (ten viruliferous thrips were released per plant at the first true leaf stage and enclosed in a cage and left for the rest of the experiment to colonize and increase their number). Thus, a total of 15 plants (5 plants per variety) and 150 thrips were released in the SVNVIT treatment cage. After 2 months, plant virus infection status was confirmed with ELISA (Agdia, Elkhart, IN, USA). Seeds from these plants were harvested on 30 May 2019.

#### 2.3.3. Plant Physiological Parameters Measurements

The experiment consisted of 5 plants per variety and a total 15 plants per treatment. Photosynthesis rate, stomatal conductance, and transpiration were measured with a LICOR 6400 equipped with IRGA (infra-red gas analyzer) (LICOR Biosciences, Lincoln, NE, USA) at monthly intervals on the fully expanded 3rd leaf from the top or upper canopy leaves [33].

#### 2.3.4. Plant Morphological Characters

Number of nodes per plant, leaf area, number of pods per plant, number of seeds per plant, and number of seeds per pod were recorded at maturity for all plants. The leaf area was measured through scanning the leaves at three positions (upper, middle, and lower plant canopy) while they were still attached to the plants. Then, the image size was determined through Image J software (ImageJ 1.51k) in Fiji.

### 2.4. Statistical Data Analysis

Data were analyzed using R version 3.5.3 [34]. Data were checked for normality through residual plotting. For estimation of average thrips number per variety during both years 2016–2017, the data were analyzed through one-way RCBD ANOVA. Bar graphs were developed using ggplot 2 package in R3.5.3. Average thrips population per year was plotted using line graphs for both years. The correlation and linear regression analysis were conducted and the results were plotted using the scatter plot package in ggplot2. The seed qualitative factors, protein, oil, carbohydrate, and fiber content, regression and correlation analysis were conducted. The seed qualitative parameter results were analyzed for different varieties through multiway ANOVA. The plant physiology lab experiment results were statistically analyzed using multi-way analysis of variance. Interaction effects were determined through interaction plotting. Diagnostic plots were checked for homogeneity of variance. Multiple comparisons were evaluated using the package “agricolae” in R 3.5.3. Tukey’s honestly significant difference was used to compare the individual means. Letters were used to rank the groups. The results were considered significant if the P values were less than 0.05. The Tukey’s test at 5% level of significance was used to establish statistical ranks. Graphs were plotted using ggplot2 [35].

## 3. Results

### 3.1. Population Abundance of Thrips in the Soybean Cultivars during 2016–2017 Field Seasons

The cumulative mean incidence of soybean thrips was significantly different among varieties and years (2016–2017) (Figure 1). The varieties which had a higher population of thrips also had the higher virus incidence (Figure 2). In both years, variety SG3322 (V1) had a significantly lower number of thrips. Hubner 3917R2x (V6) had the highest, and varieties GrowMark FS (Hisoy HS39T60) (V2), GrowMark FS Hisoy HS30A-42 (V3), Mycogen 5N343R2 (V4), Hubner H34-12R2 (V5), Syngenta S27-J7 (V7), Seedway SG3555 (V8), and Mycogen 5N312R2 (V9) had intermediate numbers (Figure 1). In 2016, the one-way ANOVA results were *p* < 0.001, DF = 9, and F = 6.816 (Figure 1). In 2017, the one-way ANOVA results were *p* < 0.001, DF = 9, and F = 6.6716 (Figure 1).

Sampling was carried out each week to determine SVNV incidence from each variety, but ELISA results were negative until August, after which the plants developed symptoms. The symptomatic leaves were SVNV ELISA-positive in 2017; however, during 2016, although sampling was carried out, the ELISA results were negative. The varieties which had the higher thrips number also had higher virus titers through ELISA (Figure 2).

The leaf area was significantly different among the varieties (*p* < 0.05, F = 2.1078, DF = 9) (Figure 3A). Leaf area at different niches (upper, middle, and lower leaves) was also significantly different (*p* < 0.001, F = 90.1077, DF = 2). The middle canopy leaves were larger than the upper and lower canopy leaves. Interaction of leaf area × leaf position was significantly different (*p* <0.001, F = 5.4927, DF = 18) (Figure 3A). Number of thrips varied significantly by variety (*p* < 0.001, F = 6.6844, DF = 9) (Figure 3B). The niche preference among varieties’ sub plot leaf position effect was also significantly different (*p* < 0.001, F = 14.5379, DF = 2). The interaction of variety, thrips, and the leaf position was non-significantly different (*p* = 0.202, F = 1.2649, DF = 18).

The number of thrips were different across different niches within the soybean plant (Figure 3B). The thrips distribution also changed with plant age (Figure 3C). During the initial stages of vegetative growth, thrips were observed across the entire plant; in later plant growth stages, the adults were predominant on new leaves and then moved towards the bean pods, and then they migrated to the later-planted beans.

The population dynamics of thrips in 2016 and 2017 were different across different months of the soybean field season (Figure 3C). Thrips populations were observed for the first time in the season at V3 stage (vegetative stage 3 is the stage of soybean plant growth when the plant has three true trifoliate leaves) on soybean plants on 8 July 2017. Thrips populations reached a peak in the month of August during both years; however, in 2017, the peak was observed in the 2nd week of August, while in 2016, it was observed in the 3rd week of August. Overall, thrips were most abundant in the month of August; the populations declined in September and October as the crop progressed towards maturity.

### 3.2. Effect of Weather Factors on Population Abundance of Soybean Thrips during 2016–2017

The correlation was non-significant and positive with air temperature, relative humidity, and rain fall, while negative correlation was observed with the wind speed and solar radiation. The populations of thrips were determined throughout the soybean growing seasons. A regression plot was developed between the weather factors (solar radiation, relative humidity, rainfall, air temperature, and wind speed) and number of thrips (Figure 4). Wind speed and thrips abundance correlation was negative in both years. The coefficient of correlation was −0.64 and −0.054. The correlation was significant in 2016 and non-significant in 2017 (*p* = 0.033, *p* = 0.88) (Figure 4E). The estimate of coefficient of regression of wind speed and thrips was negative (−0.19) and non-significant (*p* = 0.18). The regression equation was Y = 0.721 − 0.19 (wind speed) + 0.088 (year) (R^2^ = 0.1232; F = 1.335; *p* = 0.286).

### 3.3. Thrips Species on Soybeans and Nearby Crops in PA in 2018

The thrips species greatly varied in soybeans, the weeds in soybeans, melon, poenies, squash, onions, petunia, white aster daisy, and viburnum. The vector species of SVNV (*N. variabilis*, *F. tritici* and *F. fusca*) were observed on soybeans, weeds in soybeans fields, and melon. However, *F. tritici* was predominantly present on most of the surveyed plants (Table 1).

### 3.4. Seed Analysis Results

Protein content of seeds was significantly different among soybean varieties (*p* < 0.05, F = 2.5021, DF = 9) (Figure 5).

Variety Hubner H34-12R2 (V5) had the highest protein content (37.025%) and comparatively lower thrips (0.0746). However, Syngenta NK36Y6 (V10) had the lowest number of thrips and the lowest protein content; although protein content can decrease due to thrips abundance, protein content is also a genetic character that can differ across soybean varieties. A few varieties (viz., Growmark FS Hisoy HS39T60 (V2), Growmark FS Hisoy HS30A-42 (V3) and Mycogen 5N312R2 (V9) had intermediate thrips abundance and intermediate protein content. The correlation and regression of protein content and thrips abundance (Figure 6A) was non-significant and negative (*p* > 0.05). The coefficient of correlation was negative (−0.145), and the coefficient of regression was 0.02. The regression equation was Y = 37 − 0.67X.

The oil content was not significantly different in soybean cultivars (*p* > 0.05, DF = 9, F = 1.8678) (Figure 7). Although oil content is a genetic characteristic, environmental factors (such as external feeding of sap-sucking pests like thrips) can reduce plant vigor and ultimately affect the oil content. Channel3917R2X (V6) had a non-significantly higher number of thrips and an oil content of 12.05%, while Seedway SG3322 (V1) and Syngenta NKS36Y6 (V10) had non-significantly lower numbers of thrips and higher oil content (13.7%) Although the oil content was not different in varieties, the correlation of oil content with thrips abundance was significantly different and negative (*p* < 0.05). The coefficient of correlation was −0.362, the coefficient of determination was 0.13 (Figure 6B), and the regression equation was Y= 14 − 1.7X.

Carbohydrate content was significantly different across soybean cultivars (*p* < 0.001, DF = 9, F = 8.0158) (Figure 8). Although carbohydrate content is a genetic trait that differs depending on the variety, it can also be influenced by environmental factors, including thrips. The one-way ANOVA between thrips and carbohydrate content was not significant (*p* > 0.05, F = 0.866, DF = 1). The regression and correlation of thrips and carbohydrate content was non-significant (*p* > 0.05) and negative, with the regression equation Y = 19 − 0.32X. The coefficient of determination was 0.023 (Figure 6D).

Fiber content (%) was significantly different in soybean cultivars (*p* < 0.001, DF = 9, F = 13.019) (Figure 9). Although the fiber content is another genetically determined character, the varieties which had intermediate or lower numbers of thrips had the highest fiber content. For example, varieties Seedway SG3322 (V1), Growmark FS Hisoy HS39T60 (V2), Growmark FS Hisoy HS30A-42 (V3), and Mycogen 5N312R2 (V9) had lower or intermediate numbers of thrips and had higher fiber content. Variety Channel3917R2X (V6) had the highest number of thrips and had intermediate ranking of fiber content ranked ab. A one-way ANOVA comparing thrips and fiber content was non-significant. However, the regression was non-significant and negative. The regression equation was Y = 37 − 1.8X (Figure 6C).

### 3.5. Effect of SVNV on Plant Growth Parameters including Plant Physiological Parameters

#### 3.5.1. Confirmation of Virus Inoculation in the Inoculated and Infected Plants

For the physiology experiment, SVNV virus-inoculated plants inoculated through mechanical inoculation were confirmed to be positive through ELISA one month after the inoculation (Appendix A). Photographs of different treatments were taken at the R1 stage (Appendix A). The plants that showed a three-fold higher optical density (OD) value compared to the negative controls (un inoculated plants) were considered virus positive (Appendix A). The results are provided in Appendix A.

#### 3.5.2. Plant Morphological Characters

Leaf area was significantly different among treatments (*p* < 0.01, F = 43.289, DF = 2). Uninfected, untreated, undamaged control and mock plants had a larger leaf area compared to the infected and virus inoculated plants. Sub-treatment (variety) was also significantly different (*p* < 0.01, F = 9.2582, DF = 2) (Figure 10A). Interaction (treatment × variety) was non-significant (*p* > 0.05, F = 1.542, DF = 6).

The number of Internodes was significantly higher in the mock and uninfected untreated control plants (*p* < 0.01; F = 17.0571; DF = 3). However, there was no significant difference in varieties (*p* > 0.05; F = 1.3108; DF = 2). The variety and treatment interaction was non-significant (*p* > 0.05, F = 1.6202, DF = 6) (Figure 10B). The number of internodes was significantly lower in the thrips- and virus-infected plants (Figure 10B); the inoculated plants had higher numbers of internodes than the thrips- and virus-infected plants.

The number of pods was significantly lower in the SVNV infected via mechanical inoculation and SVNV infected via thrips transmission and thrips infestation compared to mock and uninfected untreated control plants (*p* < 0.01, F = 9.56, DF = 3) (Figure 11A). However, there was no significant difference in the number of pods in varieties (*p* > 0.05, F = 2.92, DF = 2). The interaction (variety × treatment) was significantly different (*p* < 0.05, F = 2.8684, DF = 6).

The total number of seeds per plant was significantly higher in the mock, uninfected, untreated control and SVNV infected via mechanical inoculation, and was significantly lower in the SVNV infected via thrips transmission and thrips infestation (*p* < 0.01; F = 14.5285; DF = 3) (Figure 11B). There was a significant difference between the varieties as well (*p* < 0.001, F = 5.8770, DF = 2). The interaction between variety and treatment was not significant (*p* > 0.05; F = 1.4436; DF = 6).

The average number of seeds per pod was not significantly different in uninfected untreated control, mock, SVNV infected via mechanical inoculation, and SVNV infected via thrips transmission and thrips infestation plants (*p* > 0.05; F = 2.2758; DF = 3) (Figure 11 C). The sub plot variety effect was significant (*p* < 0.05; F = 4.5925; DF = 2). Interaction (treatment × variety) was not significant (*p* > 0.05; F = 0.7910; DF = 6).

#### 3.5.3. Plant Photosynthetic Rate

Overall, the rate of photosynthesis (µmol CO_2_ m^−2^ S ^−1^) was significantly reduced in the SVNV infected via thrips transmission and thrips infestation plants, while the mock and the uninfected untreated control plants had higher photosynthesis rates (Figure 12A). The plant photosynthetic rate was measured at monthly intervals (Figure 12A). Plant photosynthesis (µmol/m^2^/s) was measured in different treatments using the LICOR 6400. The means were separated by Tukey’s HSD at a 0.05 level of significance. Main plot treatment (uninfected untreated control, mock, SVNV infected via mechanical inoculation and SVNV infected via thrips transmission and thrips infestation) infected plants were significantly different (*p* < 0.01; F = 21.6521, DF = 3). Sub plots (varieties) were significantly different (*p* < 0.05, F = 4.0513, DF = 2). The interaction effect (photosynthesis × main treatment × sub plot (variety) was also significantly different (*p* < 0.01, F = 21.6521, DF = 6) (Figure 12A).

The mechanical inoculation of the virus was carried out on 14 November, and the first photosynthetic measurements were taken on 16 November. At that time, the photosynthetic rate was higher in the uninfected untreated control and mock plants (Figure 13A), but it was slightly lower in the SVNV infected via thrips transmission and thrips infestation plants and lowest in the SVNV infected via mechanical inoculation plants.

After a one-month interval on 14 December, the rate of photosynthesis was equal in the uninfected untreated control, mock, and SVNV infected via mechanical inoculation plants, but lower in the SVNV infected via thrips transmission and thrips infestation plants. On 28 January, the rate of photosynthesis was lowest in the SVNV infected via thrips transmission and thrips infestation plants and inoculated plants, while it was highest in mock and uninfected, untreated control plants. On 21 February, the rate of photosynthesis was highest in the uninfected untreated control and mock plants while it was lowest in the SVNV infected via mechanical inoculation and SVNV infected via thrips transmission and thrips infestation plants (Figure 12B).

#### 3.5.4. Stomatal Conductance

The stomatal conductance rate (mol H_2_O m^−2^s^−1^) was highest in the uninfected untreated control and mock plants and lowest in the SVNV infected via mechanical inoculation, and equal in the SVNV infected via thrips transmission and thrips infestation plants (Figure 12B). The mean stomatal conductance was higher in the mock and uninfected untreated control plants on 16 November, but it was lower in SVNV infected via thrips transmission and thrips infestation plants and significantly lower in the SVNV infected via mechanical inoculation plants. The main plot treatments (uninfected, untreated control, mock, SVNV infected via mechanical inoculation, and SVNV infected via thrips transmission and thrips infestation) were significantly different (*p* < 0.01; F = 7.5048, DF = 3). Sub plots (varieties) were also significantly different (*p* < 0.05, F = 4.260, DF = 2). Finally, the interaction effect (stomatal conductance × main treatment × sub plot (variety)) was also significantly different (*p* < 0.01, F = 2.954, DF = 6) (Figure 12B).

On 14 December, the stomatal conductance was about the same in the uninfected untreated control, mock and SVNV infected via thrips transmission and thrips infestation plants but it was lowest in the SVNV infected via mechanical inoculation plants. On 28 January, the stomatal conductance was lowest in the SVNV infected via thrips transmission and thrips infestation plants, and was highest in SVNV infected via mechanical inoculation, mock, and uninfected/untreated/undamaged control plants. On 21 February, the stomatal conductance was highest in the uninfected/untreated/undamaged control plants and was lowest in the SVNV infected via thrips transmission and thrips infestation plants, while it was in between in mock and SVNV infected via mechanical inoculation plants. Overall, the stomatal conductance was lowered in the SVNV infected via mechanical inoculation and SVNV infected via thrips transmission and thrips infestation plants, and it was highest in the uninfected/untreated/undamaged control and mock plants.

#### 3.5.5. Intercellular Carbon Dioxide Content

Overall, the intercellular carbon dioxide content was equal/not significantly different in all treatments. The main plot treatments (untreated, uninfected control, mock, SVNV infected via mechanical inoculation, and SVNV infected via thrips transmission and thrips infestation) plants were not significantly different (*p* > 0.05; F = 2.27, DF = 3). Sub plots (varieties) were not significantly different (*p* > 0.05, F = 0.359, DF = 2), whereas interaction effect (intercellular carbon dioxide × main treatment × sub plot (variety)) was significantly different (*p* < 0.01, F = 6.625, DF = 6). However, intercellular carbon dioxide content is indicative of carbon dioxide entering through the stomata and the water leaves during this photosynthesis process. The intercellular carbon dioxide content was initially higher in the SVNV infected via mechanical inoculation plants, but later on it was considerably lowered in the month of December in SVNV infected via mechanical inoculated plants, which means the initial response of the SVNVMI inoculated plants had reduced stomatal opening and relied on the intercellular carbon dioxide produced during respiration, but later on it increased. But, in the SVNV infected via thrips transmission and thrips infestation plants the intercellular carbon dioxide content decreased extensively.

#### 3.5.6. Transpiration

Overall, the transpiration rate (mol m^−2^s^−1^) was significantly lower in the SVNV infected via mechanical inoculation plants and highest in the uninfected/untreated control plants and mock plants (Figure 12D). Overall, the transpiration was highest in the uninfected/untreated control and mock plants and lowest in the mechanical inoculated and SVNV infected via thrips transmission and infestation plants (Figure 13D).

The transpiration was determined in the uninfected/untreated control, SVNV infected via mechanical inoculation, mock SVNV infected via thrips transmission, and infestation plants through LICOR 6400. The means are separated by Tukey’s HSD at a 0.05 level of significance. The main plot treatments (UUCT, mock, SVNVMI, and SVNVIT plants) were significantly different (*p* < 0.05; F = 9.1230, DF = 3), whereas sub plots (varieties) were not significantly different (*p* > 0.05, F = 2.2685, DF = 2). The interaction effect (transpiration × main treatment × sub plot (variety)) was significantly different (*p* > 0.05, F = 1.6468, DF = 6) (Figure 12D).

## 4. Discussion

We investigated the composition and structure of the thrips population on soybeans, ornamentals, field crops, and weeds in central Pennsylvania. Overall, we found that vector species of SVNV *N. variabilis*, *F. fusca*, and *F. tritici* were present on soybean crops in Pennsylvania. Other thrips, *F. schultzei*, *Thrips tabaci*, and *F. occidentalis*, were also observed. Almost all of the vector species of SVNV were present on the weeds in the soybean fields. Overall, *F. tritici* was the most abundant thrips species on all crops.

Our results were similar to Chellemi et al.'s [22,36], who found that *F. tritici* was abundant in the flowering plants near soybeans and tomato fields. *F. tritici* can transmit SVNV with low efficiency. We found that *F. tritici* was present on squash, petunia, red clover, nasturtium, viburnum, and weeds. Although melon, clover, and certain weeds species have been shown to serve as the inoculum reservoir of SVNV [7,20,37], the identities of the other plants as alternative hosts of vectors have not been established yet.

*F. tritici* is abundant in greenhouses and may be an inoculum reservoir in alternative host plants. The probability of the summer weeds serving as an inoculum reservoir to shift the virus to the winter weeds is very low in Pennsylvania, as the weather is very cold, with the frost period lasting around 5 months. During this time, the above-ground parts of weeds do not survive but some parts below ground can survive; hence, the virus can replicate inside *F. tritici*, *F. fusca*, and the below-ground parts of perennial weeds. It may be possible that these thrips species do not migrate and overwinter locally as adults [36], and when the spring approaches, they emerge from the soil with the virus and transfer it first to the spring weeds. Then, when *N. variabilis* populations develop, these thrips pick up the virus from the diseased seed-borne plants and the spring weeds, and a complex of *N. variabilis*, *F. fusca***,** and *F. tritici* cause a secondary spread of the virus to the soybean crop.

Irwin et al. [38] observed thrips in Urbana, Illinois, Lexington, Kentucky, and Columbia, Missouri, and concluded that *N. variabilis* comprise 50% of thrips fauna. Our results on species composition and structure were similar to Irwin et al.'s [38], who found that *F. fusca* and *F. tritici* were dominant species. Irwin et al. (1979) also found *Leptothrips mali* (Fetch), *Dendrothrips ornatus* (Jablon-owski), *Aeolothrips bicolor*, *A. fasciatus*, and *Thrips physapus* L. to also be present. However, thrips species’ composition in our results were different from the Midwestern U.S. locations of Lexington, KY, and Urbana, IL soybean thrips fauna, where only *F. tritici*, *S. variabilis*, *A. bicolor*, and *A. fasciatus* were present. Bloomingdale et al. [22] observed thrips species in Wisconsin and Iowa by using yellow sticky traps and found *Aeolothrips*, *Anaphothrips obscurus*, *Chirothrips manicatus*, *F. fusca*, *F. occidentalis*, *F. tritici*, *F. williamsi*, *L. cerealium*, *M. abdominalis*, *N. variabilis*, and *T. tabaci* on soybeans. We did not find *C. manicatus*, *F. williamsi*, *L. cerealium*, or *M. abdominalis* in our soybean fields. Yellow sticky traps can attract thrips across a wide area, so it may be possible that the thrips inhabiting nearby crops were also captured in Bloomingdale et al. [22]. Species structure and composition may change with climate and elevation; the thrips species present on soybean in Wisconsin might be different from those present in Pennsylvania.

Thrips composition and structure may be affected by weather conditions, as well as crop phenology [39,40]. We observed the dynamics of soybean thrips populations for two years (2016 and 2017). We planted soybeans at the end of May, observed crops emerging in June, and took the first observation at the V3 stage in the field on 8 July each year. At those times, the populations of *N. variabilis* were low, but other species (viz., *F. tritici*, *F. fusca* and *F. occidentaltis*) were present. Thrips populations increased with time and reached their peaks in the third week of August in 2016 and second week of August in 2017 (Figure 3C). There were changes in weather conditions across both years (Figure 4A–E). In 2016, although the air temperature was higher in July and August (21–23 °C), it was considerably lower in September (19–21 °C). In 2017, the air temperature was lower (20–23 °C), with frequent rainfall and heavy floods (Figure 4B,D). The relative humidity in 2017 was considerably higher (81–90%) (Figure 4C).

In our results, we found a negative correlation between thrips abundance and wind speed and solar UV radiation (Figure 4A,E). Although the correlation between thrips abundance and solar UV radiation was non-significant, it showed the overall relation between two factors. We also found a positive correlation with rainfall and relative humidity (Figure 4C,D). We hypothesize that strong frequent winds may reduce the pest population development or movement. However, an additional study is needed to measure the canopy effect of wind on the soybean thrips establishment. The correlation with temperature was non-significant (Figure 4B), although temperature was higher during the crop vegetative and flowering stage until R2 stage, it decreased in September. Thrips populations may have declined due to approaching winter temperatures and crop maturity, as thrips feed on the younger and more succulent parts of plants. Our results were similar to Bloomingdale et al. [22] and Keough [14], who reported that the soybean thrips were positively correlated with degree days. We did not find any overwintering soybean thrips during soil sampling to accept Bloomingdale et al.'s [22] assumption that the soybean thrips in the north do not migrate south in the winter and instead overwinter in the north on perennials. However, there are different possibilities of thrips survival in central Pennsylvania: (1) *N. variabilis* die due to cold winter conditions of Pennsylvania and recolonize the state every year via migration from the south. (2) Soybean thrips overwinter in the soil or any other vegetation or perennial plants. (3) Soybean thrips migrate back south. Extensive studies are needed to prove the reason for thrips’ survival.

Thrips also affect bean quality. Soybean seed storage protein content usually varies between 34 to 36% [41], However, industry demands 47.5 to 48% crude protein [41]. Feed mills purchase soybean seeds based on the protein content to estimate the value of the meal [41]. During our experiment, we found that varieties vary in their protein content. When we performed ANOVA, the varieties which had the highest thrips populations also had the lowest protein content. However, there were some varieties which had lower thrips numbers but also had lower protein contents. Some varieties with the lowest thrips numbers had the lowest protein content. We are uncertain what may have led to these observations.

Overall, when plants are stressed at reproductive stages, either through herbivory or SVNV, the protein content increases and the oil content decreases [42]. However, in our experiment we did not find any correlation between thrips counts and protein content (Figure 6A). Protein content is related to plant physiological growth over time [43].

Oil content was not significantly different; however, the correlation of oil content with thrips number was significantly and negatively correlated (Figure 6B and Figure 7). Irizarry [19] pointed out that SVNV infection may lead to decreases in oil content. Although we found a significant negative correlation between thrips number and the oil content, this may be due to known genetic differences in oil content between different soybean varieties, and the presence of thrips could be correlated with preference for plants with a certain oil content, instead of being the cause of the oil content differences.

Carbohydrate content in the soybean cultivars was also significantly different (Figure 8 and Figure 6D). A few varieties with lower populations of thrips had the highest carbohydrate content, but the negative correlation was not significant (Figure 6D and Figure 8). Fiber content was lower in the varieties with higher thrips abundances (Figure 9 and Figure 6C). Because the correlation of fiber content and the thrips abundance was negative, it is possible that the overall health of the plant is affected by the virus and vector combination, or that the fiber content has an effect on the ability of thrips to feed and colonize plants. To see this process in detail, we used the growth chamber experiment.

We found that leaf area, number of internodes, and number of pods were similar in the mock and uninfected/untreated/undamaged control treatment, but lowest in SVNVIT plants (Figure 10 and Figure 11A). Seedway SG3555 (V8) had the lowest leaf area compared to Channel3917R2X (V6) and Seedway SG3322 (V1) (Figure 10). The SVNVIT plants had the lowest leaf area by a significant margin (Figure 10). Leaf area is an indicator of total plant photosynthesis rate [44]; plants with larger leaf areas photosynthesize at higher rates [44], and plants with higher photosynthesis rates produce higher yields [45]. Plants with higher photosynthesis rates have higher numbers of flowers and provide more nutrients to the developing pods [46]. Recent research in China has documented that leaf photosynthesis is the genetic determinant of increased yield in soybeans [46,47]. Increased photosynthesis resulted in increased plant biomass, which may have contributed to the yield [47]. However, there may be a misconception because increased photosynthesis results in increased vegetative growth and leaf area index, which may result in lodging [46]. In the field, bushy genotypes of soybeans with large numbers of pods per plant are needed to increase the plant yield [46]. However, taller plants with increased leaf area index may lodge and decrease interception of light during the R2–R5 stage, which may hinder plant photosynthesis and decrease the number of pods as well [47]. Larger pods may have more seeds, and increasing the number of seeds per plant may increase the yield.

However, there may be a negative effect of increased leaf area index and seed number [46] because plant biomass is dependent upon the leaf area index. Although higher leaf area leads to high plant biomass, the interception of the light by the leaves is reduced and lodging at R2–R5 may occur [48], which may reduce the overall yield. In soybeans, the yield is dependent upon photosynthesis at the R2–R5 stage, which is correlated with increased vegetative biomass, but drought, flower shedding, and extreme lower temperatures at the R2–R5 stage may decrease the yield. A reduced sink accumulation rate is related to the decrease in overall plant vegetative growth [49,50,51,52]. Herbivory from sucking insects, such as aphids at the R2 stage, may also reduce the yield, but the overall effect of thrips on plant photosynthetic rates has not been studied yet.

Overall, seed yield is related to the number of pods per plant; however, seed weight is not related to the number of pods or number of seeds per pod. Instead, in extreme drought the varieties with lower numbers of seeds per pod tend to develop larger seeds. Seed weight may not be related to the yield in some plants. In our experiments, we found that the number of seeds per pod was not significantly different among treatments.

Plant viruses may reduce photosynthesis through decreased chlorophyll content [53,54], plant dry matter content, and stomatal conductance [55], and increased transpiration due to high numbers of phytophagous vectors. Plant photosynthesis consists of two photosystems: photosystem I (PSI) and photosystem II (PSII). Photosystem II consists of water evolving complex proteins (D1 and D2) and different chlorophyll monomers. During their replication, plant viruses may deteriorate the water-evolving complex proteins diphenylcarbazide and hydroxylamine [54].

Plant viruses can affect plant physiological processes through manipulating plant biochemical reactions during replication, movement, and dispersal [56,57]. Some plant vital functions are also affected because of the plant immune process against the virus invasion [58]. For example, some plant viruses utilize the chlorophyll machinery for replication; hence, overall plant carbohydrate synthesis is affected [59]. Viruses manipulate the size exclusion limit of the plasmodesmata, and the translocation of carbohydrate to new photosynthetic leaves is affected [60]. Sugar supply is also affected, and plants remain stunted.

We observed decreases in photosynthesis in SVNV infected via thrips transmission and infestation (SVNVIT) plants compared to the mock and uninfected/untreated/undamaged control plants. Overall, the decrease in PSII efficiency in virus-infected plants is a complex process. Various plant processes may be involved in the reduction in the PSII efficiency in the virus-infected plants. In SVNV, the minor leaf veins become necrotic, and the necrotic region enlarges until eventually all of the veins in the leaf become necrotic. This necrosis can happen as a result of the plant immune system response [61]. After the infection of some viruses in a host plant cell, plasmolysis takes place, and then the cytoplasm and thylakoid membrane disintegrate and the number of mitochondria increases. The non-necrotic region continues to metabolize, while the infected cell stops working or dies. The permeability of thylakoid membranes is affected, which results in an oxidative burst and increase in peroxidase and lipoxygenase activity [62,63]. Overall, as a result of the virus infection in the plants, the plants’ photosynthetic processes can be altered. Overall, in our studies the number of internodes, average seed per pod, and total number of seeds per plant are reduced in the virus- and thrips-infected plants.

In the present study, we observed extreme yield loss in the SVNV infected via thrips transmission and infestation (SVNVIT) plants. SVNV infected via thrips transmission and infestation (SVNVIT) plants were stunted and had reduced seed number as compared to the uninfected/untreated/undamaged control (UUCT) plants.

Overall, we found that soybean thrips and other vector species are predominant on the soybean, field crops, ornamentals, and weeds. Soybean thrips were not observed during winter in the soil. Soybean thrips might travel from the Midwest to Pennsylvania, USA. Soybean varieties varied in resistance/susceptibility to vectors and viruses, but a few varieties, such as Hubner3917R2X, were highly preferred, while SwaySG3322 was non-preferred by thrips. The soybean thrips population reached a peak in the second week of August during 2016, and third week of August during 2017. Soybean thrips were predominant on upper canopy leaves. Wind speed and solar radiation were negatively correlated with thrips abundance, while air temperature, relative humidity, and rain fall were positively correlated with thrips. Oil content was significantly negatively correlated with thrips abundance. Protein, fiber, and carbohydrate content was non-significantly negatively correlated with thrips. Leaf area and internode length were significantly lower in the virus-inoculated and SVNV and thrips-infested plants. The number of seeds per plant was drastically reduced in thrips- and virus-infected plants. Thrips significantly lowered the plant yield. Plant physiological growth parameters were drastically affected by virus and vector pairing. We conclude that further studies should be carried out for developing tools to enhance the resistance of soybean cultivars

## 5. Conclusions

In light of the present study, we conclude that in Pennsylvania, the vector thrips species are present on soybeans. The soybean thrips population reaches at its peak in the month of August in central Pennsylvania. Soybean varieties vary in their attractiveness for vector population development. Severe SVNV infection and thrips colonization alters plant photosynthesis, stomatal conductance, and transpiration. These plants also produce fewer soybean seeds per plant. The heavy infestation of thrips and viruses may drastically reduce the yield by decreasing plant quality, reducing seed and pod count, and by killing plants.

## Figures and Tables

**Figure 1 viruses-15-01766-f001:**
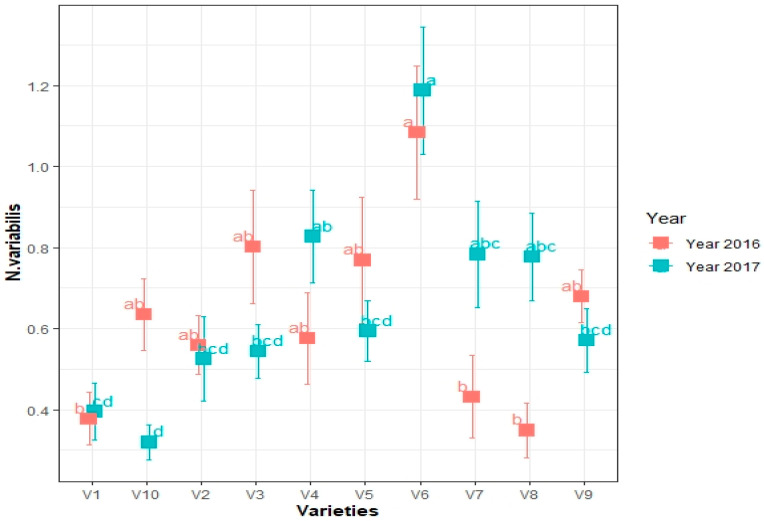
Cumulative mean incidence of soybean thrips (*N. variabilis* Beach) in different soybean cultivars during the 2016–2017 field seasons. Means are separated through Tukey’s HSD at the 0.05 level of significance. In 2016 and 2017, the population of thrips on different species was significantly different. In 2016, the one-way ANOVA results were *p* < 0.001, DF = 518.4, DF = 9, and F = 6.6816. In 2017, the one-way ANOVA results of the statistics were *p* < 0.001, DF = 9, and F = 6.6716. Here V1 = Sway SG3322, V2 = GrowMark FS Hisoy HS39T60, V3 = Grow Mark FS Hisoy HS30A-42, V4 = Mycogen 5N343R2, V5 = H3h-12R2, V6 = Hubner3917R2x, V7 = Syngenta S27-J7, V8 = Seed way SG3555, V9 = Mycogen 5N312R2, and V10 = Syngenta NKS36Y6. Here the letters represents statistical ranks based on tukey highly significant difference.

**Figure 2 viruses-15-01766-f002:**
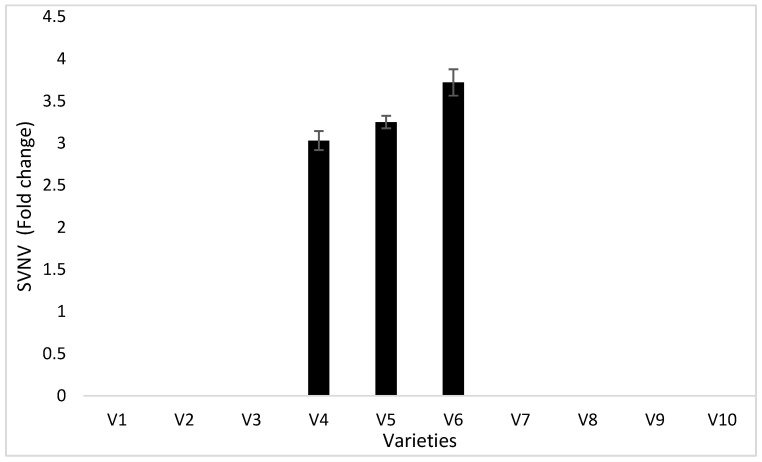
Fold change in SVNV titers in different cultivars of soybean during 2017. Here V1 = Sway SG3322, V2 = GrowMark FS Hisoy HS39T60, V3 = Grow Mark FS Hisoy HS30A-42, V4 = Mycogen 5N343R2, V5 = H3h-12R2, V6 = Hubner3917R2x, V7 = Syngenta S27-J7, V8 = Seed way SG3555, V9 = Mycogen 5N312R2, and V10 = Syngenta NKS36Y6. Here, the virus results are fold change in virus titers in leaf samples through ELISA.

**Figure 3 viruses-15-01766-f003:**
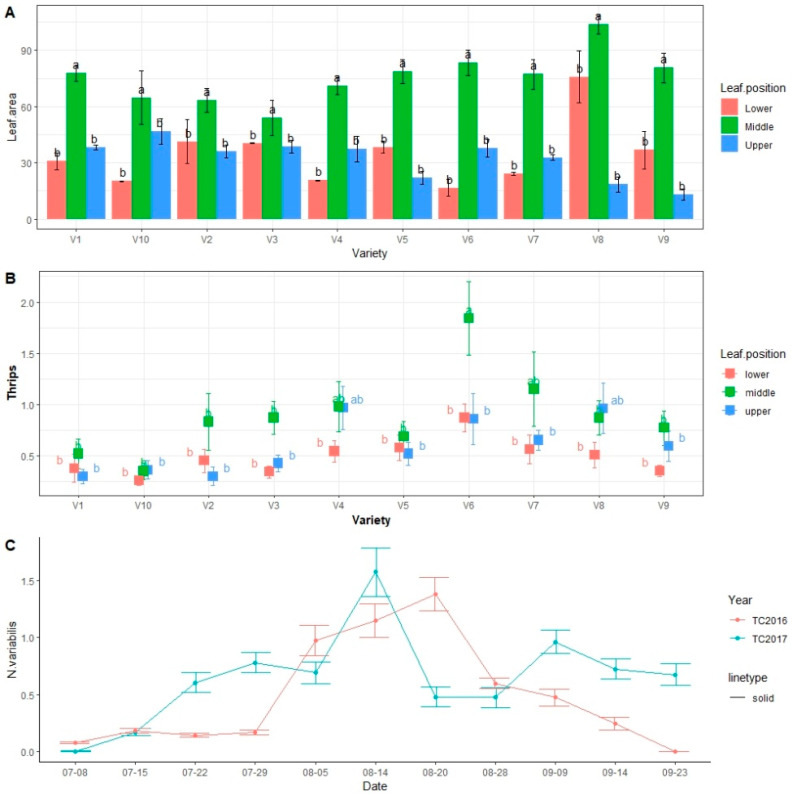
Thrips populations in different leaf niches, showing leaf area and the population fluctuation of thrips across different plant niches. (**A**) Leaf area of soybeans on different plant leaf niches (upper, middle, and lower leaves). Leaf area of different plant varieties at different leaf niches was significantly different. *p* < 0.05, F = 2.1078, DF = 9. Leaf area at different niches (upper, middle, and lower leaves) was significantly different. *p* < 0.001, F = 90.1077, DF = 2. Interaction leaf area × leaf position was significantly different *p* < 0.001, F = 5.4927, DF = 18. (**B**) Niche number of soybean thrips on different plant niches upper, middle and lower leaves. The means were separated by Tukey’s HSD at 0.05 level of significance. Main plot treatment was variety, and sub plot treatment was leaf position. The thrips number was significantly different at *p* < 0.001, F = 6.6844, DF = 9. The sub plot leaf position effect was also significantly different *p* <0.001, F = 14.5379, DF = 2. The interaction of variety, thrips, and the leaf position was non-significantly different *p* = 0.202, F = 1.2649, DF = 18. (**C**) Number of thrips in 2016 and 2017 in different months of the year during the soybean field seasons. Here, the error bars represent the standard error. Here, V1 = Sway SG3322, V2 = GrowMark FS Hisoy HS39T60, V3 = Grow Mark FS Hisoy HS30A-42, V4 = Mycogen 5N343R2, V5 = H3h-12R2, V6 = Hubner3917R2x, V7 = Syngenta S27-J7, V8 = Seed way SG3555, V9 = Mycogen 5N312R2, and V10 = Syngenta NKS36Y6. Here the letters represent significant difference among treatments based on Tukey highly significant difference (Tukey HSD).

**Figure 4 viruses-15-01766-f004:**
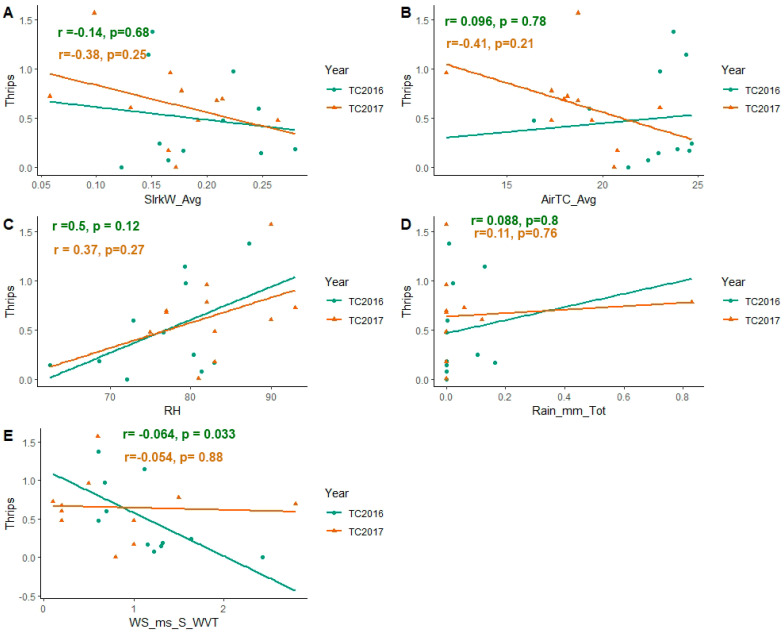
Scatter regression plots between the weather factors and the thrips number in 2016 and 2017. (**A**) Scattered regression plot between solar radiation and thrips abundance. The coefficient of correlation in both years was negative (R = −0.14 and −0.38) and there was no significant correlation of the solar radiation with the average number of thrips per plant leaf (*p* = 0.68 & 0.25). (**B**) Scattered correlation and regression plot of air temperature (C°) to thrips abundance. The coefficient of correlation was non-significant. (**C**) Scatter-plot correlation and regression plot between relative humidity and thrips was a non-significant (*p* = 0.12, *p* = 0.27) (**D**) The correlation and regression scatter-plot between rainfall and thrips abundance The coefficient of correlation was non-significant for both years. (**E**) The scatter-plot of correlation and regression between the speed of wind and the thrips abundance was significant (*p* = 0.88, *p* = 0.03) and negative. Here TC2016 and TC2016 means thrips counts during 2016 and 2017. Here, SlrkW_Avg = average solar radiation; AirTC_Avg = average air temperature; RH = relative humidity; Rain_mm_Tot = total rain fall in mm; WS_ms_S_WVT = wind speed.

**Figure 5 viruses-15-01766-f005:**
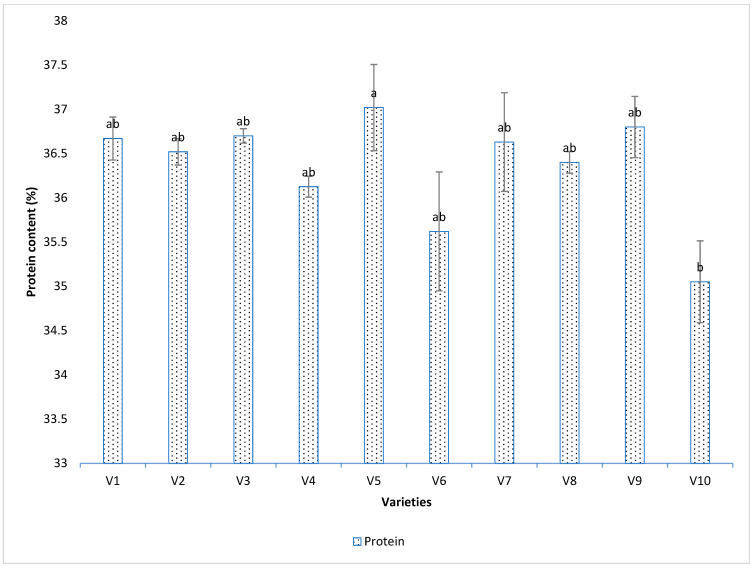
Percent protein content in different soybean varieties. Varieties were coded V1–V10. Means were separated by Tukey’s HSD at 0.05 level of significance. Protein content was significantly different (*p* < 0.05, F = 2.5021, DF = 9) in all varieties. Here, V1 = Sway SG3322, V2 = GrowMark FS Hisoy HS39T60, V3 = Grow Mark FS Hisoy HS30A-42, V4 = Mycogen 5N343R2, V5 = H3h-12R2, V6 = Hubner3917R2x, V7 = Syngenta S27-J7, V8 = Seed way SG3555, V9 = Mycogen 5N312R2, and V10 = Syngenta NKS36Y6. Here the letters represent statistical difference among treatments based on Tukey Highly significant difference.

**Figure 6 viruses-15-01766-f006:**
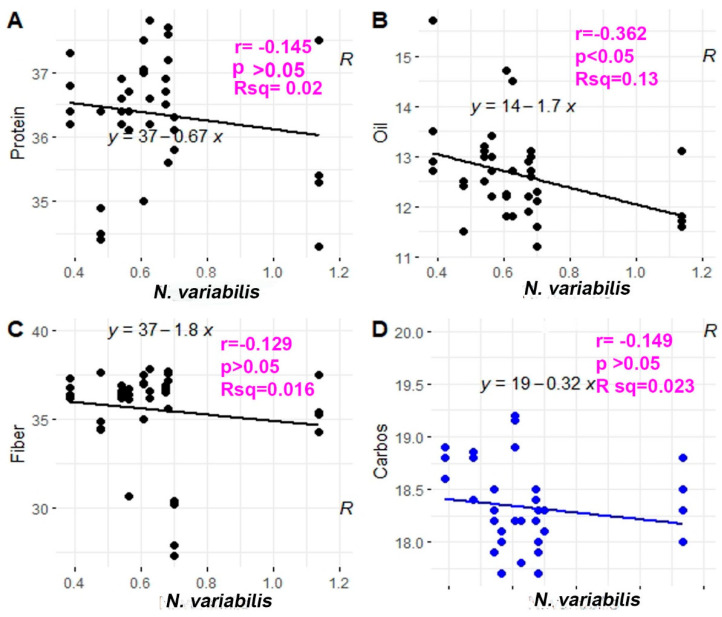
Regression and correlation plots in % protein, % oil, % fiber, and % carbohydrates against soybean thrips (*N. variabilis*). Thrips population was observed on different varieties during the 2016 and 2017 field seasons. The regression is plotted in between thrips counts on ten different varieties. (**A**) Regression plot between *N. variabilis* and fiber content. (**B**) Regression plot between *N. variabilis* and protein content. (**C**) Regression plot between *N. variabilis* and oil content. (**D**) Regression plot between *N. variabilis* and carbohydrate content in soybean seeds.

**Figure 7 viruses-15-01766-f007:**
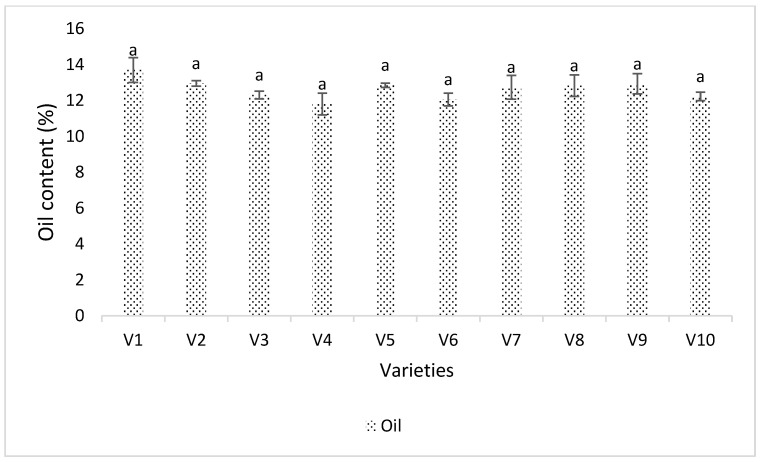
Percent oil content in soybean seeds and thrips abundance in different soybean varieties. Varieties were coded V1–V10. Means were separated by Tukey’s HSD at 0.05 level of significance. Here, V1 = Sway SG3322, V2 = GrowMark FS Hisoy HS39T60, V3 = Grow Mark FS Hisoy HS30A-42, V4 = Mycogen 5N343R2, V5 = H3h-12R2, V6 = Hubner3917R2x, V7 = Syngenta S27-J7, V8 = Seed way SG3555, V9 = Mycogen 5N312R2, and V10 = Syngenta NKS36Y6. Here the letters represent difference among treatments based upon Tukey Highly Significant Difference among means.

**Figure 8 viruses-15-01766-f008:**
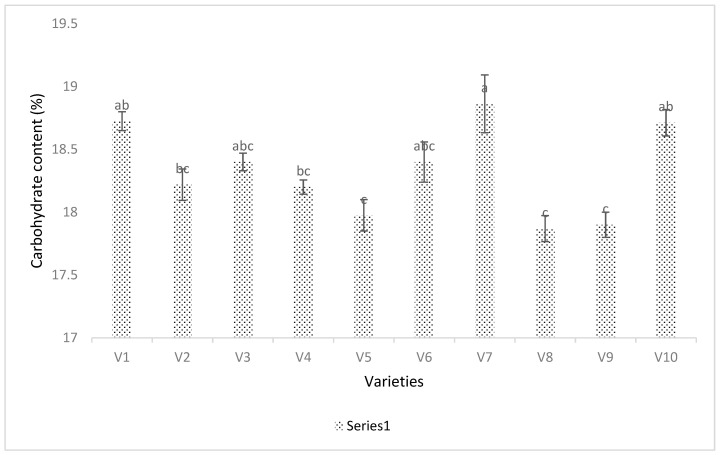
Percent carbohydrate content and thrips abundance in different soybean varieties. Varieties were coded V1–V10. Means were separated by Tukey’s HSD at a 0.05 level of significance. Thrips ranking is in black text and carbohydrate ranking is in red text. Error bars represent the standard error. Overall, there were significant differences (*p* < 0.001, DF = 9, F = 8.0158) found in the carbohydrate content of soybeans during the 2016 and 2017 field seasons. Here, V1 = Sway SG3322, V2 = GrowMark FS Hisoy HS39T60, V3 = Grow Mark FS Hisoy HS30A-42, V4 = Mycogen 5N343R2, V5 = H3h-12R2, V6 = Hubner3917R2x, V7 = Syngenta S27-J7, V8 = Seed way SG3555, V9 = Mycogen 5N312R2, and V10 = Syngenta NKS36Y6. Here the letters represent significant difference among treatments based upon Tukey Highly Significant difference test.

**Figure 9 viruses-15-01766-f009:**
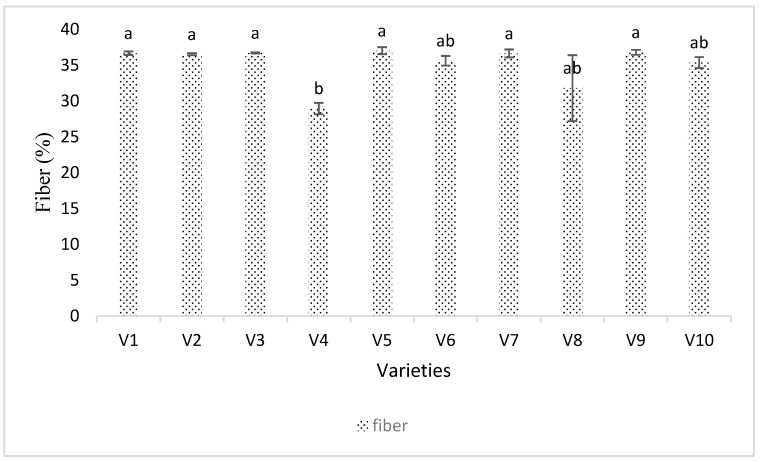
Percent fiber content and thrips abundance in different soybean varieties. Varieties were coded V1–V10. Means were separated by Tukey’s HSD at a 0.05 level of significance. Overall, there were significant differences (*p* < 0.001, DF = 9, F = 13.019) found in the fiber content number in soybeans during the 2016 and 2017 field seasons. Here, V1 = Sway SG3322, V2 = GrowMark FS Hisoy HS39T60, V3 = Grow Mark FS Hisoy HS30A-42, V4 = Mycogen 5N343R2, V5 = H3h-12R2, V6 = Hubner3917R2x, V7 = Syngenta S27-J7, V8 = Seed way SG3555, V9 = Mycogen 5N312R2, and V10 = Syngenta NKS36Y6. Here the letters represent significant difference among treatments based upon Tukey Highly Significant difference test.

**Figure 10 viruses-15-01766-f010:**
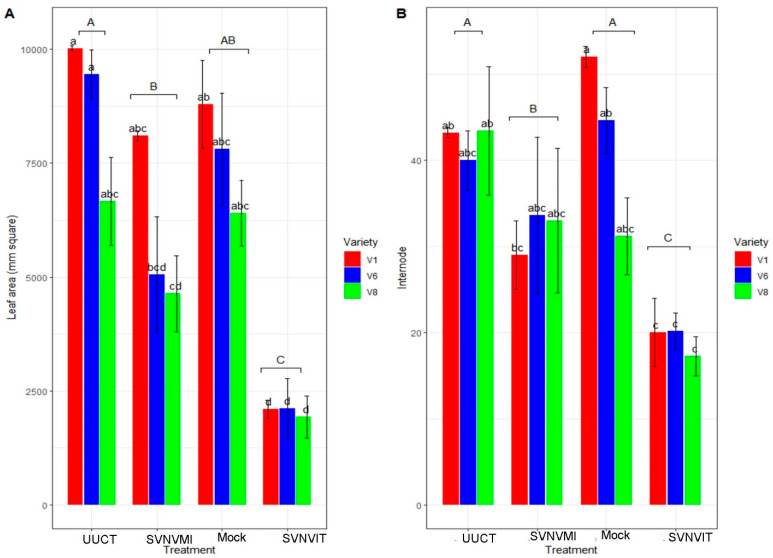
Growth chamber results. Leaf area and number of internodes in different treatments. Here, UUCT–uninfected/untreated/undamaged control treatment plants; SVNVMI—SVNV infected via mechanical inoculation; mock—buffer was rubbed on leaves at V3 stage; and SVNVIT—SVNV infected via thrips transmission and thrips infestation. (**A**) Leaf area in UUCT, SVNVMI, mock, and SVNVIT plants. Means were separated at <0.05 Tukey’s HSD. Main plot treatments—UUCT, SVNVMI, mock, and SVNVIT plants. Treatments were significantly different (*p* < 0.01, F = 43.289, DF = 2). Sub-treatment (variety) was also significantly different (*p* < 0.01, F = 9.2582, DF = 2). Interaction (treatment × variety) was non-significant (*p* > 0.05, F = 1.542, DF = 6). (**B**) Number of internodes per plant in UUCT, SVNVMI, mock, and SVNVIT plants. Means were separated at <0.05 Tukey’s HSD. Main plot treatment (UUCT, SVNVMI, mock, and SVNVIT plants) were significantly different (*p* < 0.01; F = 17.05, DF = 3). Sub plot (variety) differences were non-significant (*p* > 0.05, F = 1.3108, DF = 2). Interaction effect (Internodes length × main treatment × sub plot (variety) was also not significantly different (*p* > 0.05, F = 1.6202, DF = 6). Here V1 = Sway SG3322, V6 = Hubner3917R2x and V8 = Seed way SG3555. Here the letters represent significant difference among treatments based upon Tukey Highly Significant difference test.

**Figure 11 viruses-15-01766-f011:**
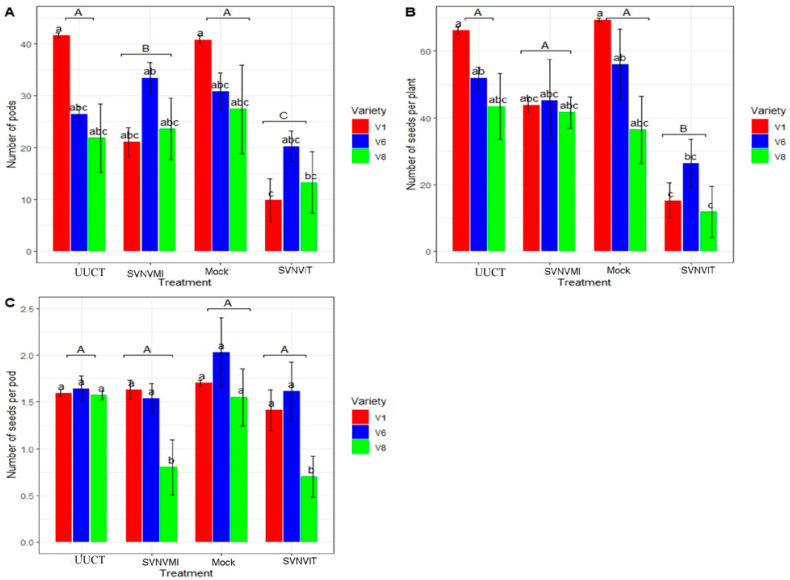
Growth chamber results. Number of pods, number of seeds per plant, and number of seeds per pod in different treatments and cultivars. Here, UUCT—uninfected/untreated/undamaged control treatment plants; SVNVMI—SVNV infected via mechanical inoculation; mock—buffer was rubbed on leaves at V3 stage; and SVNVIT—SVNV infected via thrips transmission and thrips infestation. (**A**) Number of pods per plant in UUCT, SVNVMI, mock, and SVNVIT plants. Means were separated at <0.05 Tukey’s HSD. Infected main plot treatment plants (UUCT, mock, SVNVMI, and SVNVIT) were significantly different (*p* < 0.01; F = 9.56, DF = 3). Sub plots (varieties) were not significantly different (*p* > 0.05, F = 2.92, DF = 2). Interaction effect (number of pods × main treatment × sub plot (variety)) was significantly different (*p* < 0.05, F = 2.8684, DF = 6). (**B**) Number of seeds per plant in UUCT, SVNVMI, mock, and SVNVIT plants. Means were separated at <0.05 Tukey’s HSD. Infected main plot treatment plants (UUCT, SVNVMI, mock and SVNVIT) were significantly different (*p* < 0.01; F = 9.56, DF = 3). Sub plots (varieties) were not significantly different (*p* > 0.05, F = 2.92, DF = 2). Interaction effect (number of pods × main treatment × sub plot (variety)) was significantly different (*p* < 0.05, F = 2.8684, DF = 6). (**C**) Number of seeds per pod in UUCT, SVNVMI, mock, and SVNVIT plants. Means were separated at <0.05 Tukey’s HSD. Main plot treatment (UUCT, SVNVMI, mock, and SVNVIT) plants were not significantly different (*p* > 0.05; F = 2.27, DF = 3). Sub plots (varieties) were also not significantly different (*p* > 0.05, F = 4.59, DF = 2). Finally, interaction effect (number of seeds per pod × main treatment × sub plot (variety)) was not significantly different (*p* > 0.05, F = 0.7910, DF = 6). Here, V1 = Sway SG3322, V6 = Hubner3917R2x, and V8 = Seed way SG3555. Here the letters represent significant difference among treatments based upon Tukey Highly Significant difference test.

**Figure 12 viruses-15-01766-f012:**
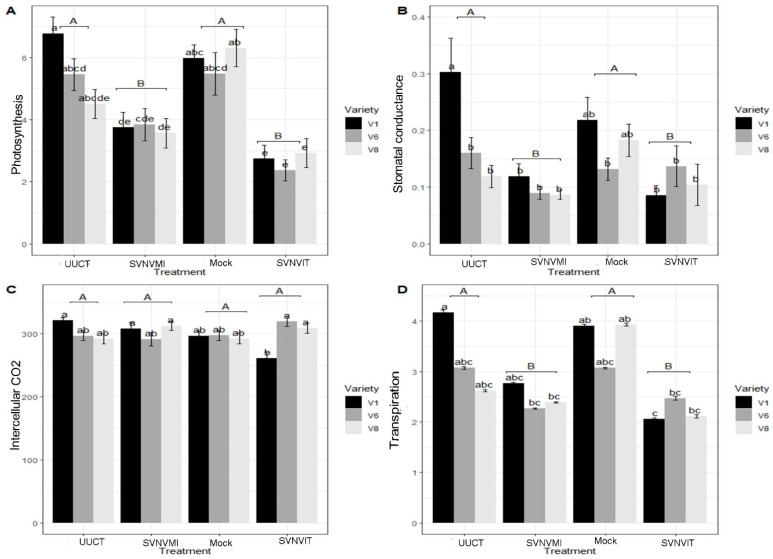
Photosynthesis, stomatal conductance, intercellular carbon dioxide, and transpiration in different treatments and cultivars. Here, UUCT—uninfected/untreated/undamaged control treatment plants; SVNVMI—SVNV infected via mechanical inoculation; mock—buffer was rubbed on leaves at V3 stage; and SVNVIT—SVNV infected via thrips transmission and thrips infestation. (**A**) Plant photosynthesis in different treatments. Means were separated at <0.05 Tukey’s HSD. Main plot treatment (UUCT, mock, SVNVMI, and SVNVIT) plants were significantly different *p* < 0.01; F = 21.6521, DF = 3. Sub plots (varieties) were significantly different. *p* < 0.05, F = 4.0513, DF = 2. Interaction effect (photosynthesis × main treatment × sub plot (variety)) was significantly different with *p* < 0.01, F = 21.6521, DF = 6. (**B**) Stomatal conductance in UUCT, mock, SVNVMI, and SVNVIT plants. Means were separated at <0.05 Tukey’s HSD. Infected main plot treatment plants (UUCT, mock, SVNVMI, and SVNVIT) were significantly different *p* < 0.01; F = 7.5048, DF = 3. Sub plots (varieties) were significantly different *p* < 0.05, F = 4.260, DF = 2. Interaction effect (stomatal conductance × main treatment × sub plot (variety)) was significantly different with *p* < 0.01, F = 2.954, DF = 6. (**C**) Intercellular carbon dioxide content in UUCT, mock, SVNVMI, and SVNVIT plants. Means were separated at <0.05 Tukey’s HSD. Main plot treatment (UUCT, mock, SVNVMI, and SVNVIT) plants were significantly different with *p* > 0.05; F = 2.27, DF = 3. Sub plots (varieties) were non-significantly different. *P* > 0.05, F = 0.359, DF = 2. Interaction effect (intercellular carbon dioxide × main treatment × sub plot (variety)) was significantly different with *p* < 0.01, F = 6.625, DF = 6. (**D**) Transpiration in UUCT, mock, SVNVMI, and SVNVIT plants. Means were separated at <0.05 Tukey’s HSD. Main plot treatment (UUCT, mock, SVNVMI, and SVNVIT) plants were significantly different, with *p* < 0.05; F = 9.1230, DF = 3. Sub plots (varieties) were non-significantly different, with *p* > 0.05, F = 2.2685, DF = 2. Interaction effect (transpiration × main treatment × sub plot (variety)) was significantly different with *p* > 0.05, F = 1.6468, DF = 6. Here, V1 = Sway SG3322, V6 = Hubner3917R2x and V8 = Seed way SG3555. Here the letters represent significant difference among treatments based upon Tukey Highly Significant difference test.

**Figure 13 viruses-15-01766-f013:**
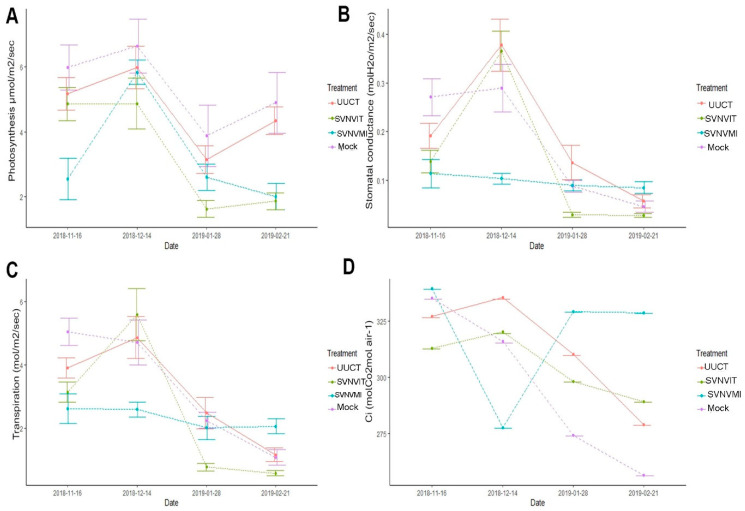
Cumulative effect of different treatments on the plant photosynthesis rate during growth of soybean plants from maturity to flowering. Here, UUCT—uninfected/untreated/undamaged control treatment plants; SVNVMI—SVNV infected via mechanical inoculation; mock—buffer was rubbed on leaves at V3 stage; SVNVIT—SVNV infected via thrips transmission and thrips infestation. Means were separated at <0.05 Tukey’s HSD. (**A**) Photosynthesis rate (µmol/m^2^/s) in the UUCT, mock, SVNVMI, and SVNVIT. (**B**) Stomatal conductance (molH_2_O/m^2^/s) in UUCT, mock, SVNVMI, and SVNVIT plants. (**C**) Transpiration rate (mol/m^2^/s) in UUCT, mock, SVNVMI, and SVNVIT plants. (**D**) Intercellular carbon dioxide (CO_2_ mol air^−1^) in the UUCT, mock, SVNVMI, and SVNVIT plants.

**Table 1 viruses-15-01766-t001:** Thrips species fauna found on soybean and nearby crops in Pennsylvania during Summer 2018.

Host Plant	Species
Soybean	*Frankliniella fusca*
Soybean	*Frankliniella* sp.
Soybean	*Neohydatothrips variabilis (*Beach*)*
Soybean	*Frankliniella schultzei*
Soybean	*Frankliniella tritici*
Soybean	*Frankliniella occidentalis*
Soybean	*Thrips tabaci*
Weeds in soybean	*Frankliniella tritici*
Weeds in soybean	*Anaphothrips obscurus*
Weeds in soybean	*Frankliniella schultzei*
Weeds in soybean	*Frankliniella* sp.
Weeds in soybean	*Neohydatothrips variabilis (*Beach*)*
Squash	*Frankliniella* sp.
Squash	*Thrips tabaci*
Squash	*Frankliniella tritici*
Melon	*Frankliniella tritici*
Melon	*Frankliniella* sp.
Melon	*Neohydatothrips variabilis*
Petunia	*Frankliniella* sp.
Petunia	*Frankliniella tritici*
Onions	*Frankliniella* sp.
Onions	*Thrips tabaci*
Peony	*Haplothrips gowdeyi*
Peony	*Frankliniella tritici*
Red clover	*Frankliniella tritici*
Nasturtium	*Frankliniella tritici*
Nasturtium	*Frankliniella fusca*
White aster daisy	*Frankliniella tritici*
Viburnum	*Frankliniella tritici*

## Data Availability

All data are present in the paper.

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
