# Peer review of "Ecological Interactions among Thrips, Soybean Plants, and Soybean Vein Necrosis Virus in Pennsylvania, USA"

_viruses, 2023, doi:10.3390/v15081766_

Round 1

Reviewer 1 Report

This manuscript surveyed the population dynamics of thrips species and disease incidence of SVNV on different plant host species over years and their impacts on grain quality. Field and greenhouse experiments were also conducted to evaluate the consequences of thrips infestation and virus infection on soybean physiology and seed quality. Overall, the study was performed properly and only a few minor comments are required to be addressed.

Line 105: provide the information about the origins of these 10 varieties.

Line 98: provide the time when soybeans were planted and harvested in the field in the methods section.

Line 134-138: correct the grammar errors present in these sentences.

Line 139: describe in more detail how the regression analysis was performed to study the relationships between thrips abundance and different environmental factors.

Line 145: confusing sentence “Grain quality was assessed to determine the impact of virus on crop quality.” If these seeds were from field plants, how did you determine that the grain quality was affected by virus only? Shouldn’t the thrips abundance factor be considered too?  

Line 150: similar issue as above. In the previous sentence, authors tried to determine the impact of virus infection on grain quality. However, in this sentence, “Grain quality parameters (oil content, carbohydrate content, and protein content) were plotted against thrips abundance”, authors indicated the relationship between grain quality and thrips abundance, which is confusing. Please clarify that.  

Line 179: indicate the age of soybean plants being inoculated.

Line 212: remove letter “I” at the end of the sentence.

Fig 2: it is unclear how the fold change of SVNV was calculated using ELISA results. Please provide more explanation. Due to the limited accuracy of ELISA to measure virus titers in samples, I highly recommend authors to show the disease incidences, instead of titers, for SVNV infection across the varieties. Showing both may be an alternative option too.

Table 1 shows the thrips species identified on different host plants (qualitative results). I recommend creating another column in this table to show the quantitative results - frequency (total/average number) of each thrips species per host plant species, which will be required to support authors’ statement “Overall, F. tritici was the most abundant thrips species on all crops.”

Fig 4: provide the corresponding p-value and correlation coefficient (r) for each plot in Fig 4.

Line 661: I suggest authors should remind readers that some correlations were not statistically significant though the trends may be informative on some level.  

To determine the infection status of mechanical inoculated plants, which leaves were collected for testing? In another words, were the infection systemic?  

Manuscript was overall well-written without major grammar issues. 

Author Response

Response to Reviewer 1 Comments

Comments and suggestion: This manuscript surveyed the population dynamics of thrips species and disease incidence of SVNV on different plant host species over years and their impacts on grain quality. Field and greenhouse experiments were also conducted to evaluate the consequences of thrips infestation and virus infection on soybean physiology and seed quality. Overall, the study was performed properly and only a few minor comments are required to be addressed. 

Response to reviewer comments: We appreciate the comments by the reviewer 1 and we think paper has been improved after incorporating these suggestions and comments.

Line 105: provide the information about the origins of these 10 varieties.

The information about origin of varieties is provided. The modified sentence is “Ten soybean varieties were planted, including Sway SG3322 (V1) (Seed way, New York USA), GrowMark FS (Hisoy HS39T60) (V2) (Growmark FS, LLC, Delaware USA), Grow Mark FS Hisoy HS30A-42 (V3) (Growmark FS, LLC, Delaware USA), Mycogen 5N343R2 (V4) (Dow Agro Sciences, Canada), H3h-12R2 (V5) (Hubner seeds, USA), Hubner3917R2x (V6) (Hubner seeds, USA), Syngenta S27-J7 (V7) (Syngenta, USA), Seed way SG3555 (V8) (Seed way, New York USA), Mycogen 5N312R2 (V9) (Dow Agro Sciences, Canada), and Syngenta NKS36Y6 (V10) (Syngenta, USA)”

Line 98: provide the time when soybeans were planted and harvested in the field in the methods section.

The soybeans were planted in Early July and harvested in November each year.

Line 134-138: correct the grammar errors present in these sentences.

The grammar error was corrected. The modified sentence is “During 2017 all soybean samples collected during the crop growth stages were negative until symptoms appearance in August, however, after appearance of symptoms, the samples taken from symptomatic plants were positive through ELISA”.

Line 139: Describe in more detail how the regression analysis was performed to study the relationships between thrips abundance and different environmental factors.

Mean thrips population was plotted against weather factors viz., temperature, rainfall, relative humidity, wind velocity and solar radiation. Regression and correlation analysis was done through R 3.5.3.

Line 145: confusing sentence “Grain quality was assessed to determine the impact of virus on crop quality.” If these seeds were from field plants, how did you determine that the grain quality was affected by virus only? Shouldn’t the thrips abundance factor be considered too? 

The sentence was modified according to reviewer suggestion. The modified sentence is “Although in the field conditions, multiple factors lower the quality of grains including insects and diseases. Multiple viruses and diseases may lower the crop quality. Hence to determine the impact of thrips and SVNV incidence on the seed quality, 500g seeds were harvested from each variety and each replicate in the field experiment (4 replicates per each variety totaling 40 samples per experiment per year) and sent to the Grain Quality Lab at Iowa State University, USA (3167 National Swine Research and Information Center (NSRIC))”.

Line 150: similar issue as above. In the previous sentence, authors tried to determine the impact of virus infection on grain quality. However, in this sentence, “Grain quality parameters (oil content, carbohydrate content, and protein content) were plotted against thrips abundance”, authors indicated the relationship between grain quality and thrips abundance, which is confusing. Please clarify that.

The sentence was clarified as above.

Line 179: indicate the age of soybean plants being inoculated.

Soybeans plants at V2 stage were selected for virus inoculation.

Line 212: remove letter “I” at the end of the sentence.

Removed

Fig 2: it is unclear how the fold change of SVNV was calculated using ELISA results. Please provide more explanation. Due to the limited accuracy of ELISA to measure virus titers in samples, I highly recommend authors to show the disease incidences, instead of titers, for SVNV infection across the varieties. Showing both may be an alternative option too.

During ELISA, we kept a standard SVNV positive, a standard negative (PBST Buffer) and the samples. The fold change in virus titers was calculated through comparing absorbance value at 450 nm in positive and samples.

Table 1 shows the thrips species identified on different host plants (qualitative results). I recommend creating another column in this table to show the quantitative results - frequency (total/average number) of each thrips species per host plant species, which will be required to support authors’ statement “Overall, F. tritici was the most abundant thrips species on all crops.”

During my studies, I collected thrips and prepared slides. I sent the thrips slides to thrips taxonomist for species confirmation. Although, I did my best to prepare good slides, however, some slides were not as good as they should be? Hence some slides were rejected. In this way the total number of thrips species identified and total number of thrips species collected during sampling is not same. Hence, I cannot make another column to show the frequency of thrips as this may be not accurate.

Fig 4: provide the corresponding p-value and correlation coefficient (r) for each plot in Fig 4.

 The corresponding p-value and correlation coefficient value was provided.

Line 661: I suggest authors should remind readers that some correlations were not statistically significant though the trends may be informative on some level. 

We provided the requisite information. The modified sentence is “In our results, we found a negative correlation between thrips abundance and wind speed and solar UV radiation (Fig. 4A & E). Although the correlation between thrips abundance and solar UV radiation was non-significant but it showed overall relation between two factors. We also found a positive correlation with rainfall and relative humidity (Fig. 4C & D).”

To determine the infection status of mechanical inoculated plants, which leaves were collected for testing? In another words, were the infection systemic? 

Soybeans plants at V2 stage were selected for virus inoculation. The requisite information is provided

Reviewer 2 Report

In the manuscript by Hameed et al., the authors conducted a study on the composition and structure of the thrips population on various crops in central Pennsylvania, including soybeans, ornamentals, field crops, and weeds. The study found that F. tritici was the most abundant thrips species on all crops. Additionally, vector species of SVNV, namely N. variabilisF. fusca, and F. tritici, were present on soybean crops in Pennsylvania. Other thrips species, such as F. schultzeiThrips tabaci, and F. occidentalis, were also observed. The study further revealed that almost all vector species of SVNV were present on the weeds in the soybean fields.

Furthermore, the authors investigated the resistance of soybean cultivars against SVNV and studied the hibernation behavior of thrips under field conditions. Field and lab experiments were conducted to determine disease incidence and vector abundance in different soybean genotypes. The impact of the virus, vector, and their combination on soybean physiology was also evaluated. Interestingly, the study showed that the plant photosynthetic rate, stomatal conductance, intercellular carbon dioxide content, and transpiration differed between SVNV-infected plants via mechanical inoculation and those infected via thrips transmission and thrips infestation.

Overall, the knowledge gained from this research could be valuable for the development of integrated pest management (IPM) strategies against thrips and the dispersal of SVNV in the field. The authors suggest accepting the manuscript after revising the resolution of figure 4.

Author Response

Response to Reviewer 2 Comments

Comments and suggestion:

In the manuscript by Hameed et al., the authors conducted a study on the composition and structure of the thrips population on various crops in central Pennsylvania, including soybeans, ornamentals, field crops, and weeds. The study found that F. tritici was the most abundant thrips species on all crops. Additionally, vector species of SVNV, namely N. variabilisF. fusca, and F. tritici, were present on soybean crops in Pennsylvania. Other thrips species, such as F. schultzeiThrips tabaci, and F. occidentalis, were also observed. The study further revealed that almost all vector species of SVNV were present on the weeds in the soybean fields.

Furthermore, the authors investigated the resistance of soybean cultivars against SVNV and studied the hibernation behavior of thrips under field conditions. Field and lab experiments were conducted to determine disease incidence and vector abundance in different soybean genotypes. The impact of the virus, vector, and their combination on soybean physiology was also evaluated. Interestingly, the study showed that the plant photosynthetic rate, stomatal conductance, intercellular carbon dioxide content, and transpiration differed between SVNV-infected plants via mechanical inoculation and those infected via thrips transmission and thrips infestation.

Overall, the knowledge gained from this research could be valuable for the development of integrated pest management (IPM) strategies against thrips and the dispersal of SVNV in the field. The authors suggest accepting the manuscript after revising the resolution of figure 4.

Response to reviewer comments: We appreciate the comments by the reviewer 2 and we think paper has been improved after incorporating these suggestions and comments. The Figure 4 has been replaced.
